# Surgical Emergencies in Patients with Hemophilia A—What to Expect

**DOI:** 10.3390/healthcare12060610

**Published:** 2024-03-07

**Authors:** Adelina Tanevski, Bogdan Mihnea Ciuntu, Oana Viola Badulescu, David Ovidiu Buescu, Mihai Marius Zuzu, Valerii Lutenco, Raul Mihailov, Ciprian Cirdeiu, Dan Vintila, Lili Gabriela Lozneanu, Dan Andronic, Stefan Octavian Georgescu

**Affiliations:** 1Department of General Surgery, Faculty of Medicine, Grigore T. Popa University of Medicine and Pharmacy, 16 Universitatii Street, 700115 Iasi, Romania; papancea.adelina@umfiasi.ro (A.T.); buescu_david-ovidiu@d.umfiasi.ro (D.O.B.); zuzumihai@gmail.com (M.M.Z.); cipc75@yahoo.com (C.C.); dan.vintila@umfiasi.ro (D.V.); lili.lozneanu@umfiasi.ro (L.G.L.); dan.andronic@umfiasi.ro (D.A.); stefan.georgescu@umfiasi.ro (S.O.G.); 2General Surgery Clinic, “St. Spiridon” County Emergency Clinical Hospital, 1 Independence Boulevard, 700111 Iasi, Romania; 3Department of Hematology, Faculty of Medicine, Grigore T. Popa University of Medicine and Pharmacy, 16 Universitatii Street, 700115 Iasi, Romania; 4Department of General Surgery, Faculty of Medicine, “Dunarea de Jos” University of Medicine and Pharmacy, 800010 Galati, Romania; vl187@student.ugal.ro (V.L.); raul.mihailov@ugal.ro (R.M.); 5General Surgery Clinic, “St. Apostol Andrei” County Emergency Clinical Hospital, Strada Brăilei 177, 800578 Galati, Romania

**Keywords:** hemophilia A, bleeding disorders, coagulation, hemostasis, surgical emergencies

## Abstract

Surgical emergencies in patients with hemophilia A represent a major risk of mortality without proper multidisciplinary management and require prompt and effective treatment to prevent complications and improve patient outcomes. We present a short number of cases that were hospitalized in the I–II Surgery Clinic of the Emergency County Hospital “St. Spiridon” from Iasi, Romania, with hemophilia A requiring surgical emergencies. The timing of surgical intervention is very important, so the indication for surgical intervention must be made judiciously and without delay. Consequently, it is vital to ensure access to hemostatic support so surgery can be performed on these patients, ultimately saving their lives.

## 1. Introduction

Hemophilia A and B are X-linked congenital deficiencies of factor VIII and factor IX, respectively, that occur in 0.001% of the population, with hemophilia A accounting for 85% of cases [1]. Hemophilia severity is classified based on factor activity level as mild (5–40%), moderate (1–5%), or severe (<1%). Patients with mild hemophilia often are asymptomatic and become coagulopathic only after massive trauma or major surgery. Patients with moderate or severe hemophilia may bleed spontaneously or with minor trauma [2]. Bleeding can develop spontaneously in patients with a severe deficiency of coagulation factors, but in cases of invasive procedures of trauma, there is an increased additional risk of intra- and postoperative hemorrhage, even with properly administered hemostatic treatment [3]. In the case of the hemophiliac patient, the ideal indication is for conservative treatment, considering the risk of bleeding, but conservative treatment cannot be instituted in cases with an acute surgical abdomen, with the only solution being emergency surgery.

In patients with hemophilia A who require surgery, it is important to assess the risk of perioperative bleeding before surgery and replace the deficient coagulation factors to ensure that the procedure can be performed safely [4]. Traditionally, surgical procedures in individuals with hemophilia are classified as either major or minor. This classification is frequently used to indicate the expected risk of bleeding associated with the surgery, with this being the main concern in carrying out the therapeutic plan [5].

We report on five patients with hemophilia A who needed surgical procedures. Out of these, four required urgent surgical intervention. Additionally, there was one case where conservative treatment was initially applied but later changed to surgical intervention due to unsatisfactory progress. In all these instances, patients with hemophilia A received appropriate substitution therapy with a coagulation factor from the time of their admission, during surgery, and in the postoperative period.

## 2. Results

### 2.1. Case Report

#### 2.1.1. Case 1

A 71-year-old female patient diagnosed with autoimmune coagulopathy secondary to Leflunomide treatment in the context of rheumatoid arthritis was admitted to the emergency unit with a tumor-like growth in the right thigh, painful, with edema in the calf and right leg, chills and fever. From a clinical and paraclinical examination (Table 1), the suspicion of a superinfected hematoma was raised.

A broad-spectrum antibiotic treatment was administered, and the surgical intervention involved performing a puncture and evacuation of 500 mL of fluid with a purulent appearance but with a negative antibiogram.

Throughout the hospital stay, multiple drainage punctures were carried out, resulting in the complete drainage of the abscess, with volumes of 700 mL, 300 mL, and 175 mL, respectively, and in the final two punctures, the fluid drained appeared serious. During the patient’s hospitalization, she received a complex of anti-inhibitors of factor VIII activity (FEIBA), with postoperative monitoring without hemorrhagic events and with the resolution of calf collection. The patient was discharged after 21 days of hospitalization.

The 6-month monitoring under treatment with eptacog alfa at 90 µg/kgbw was without hemorrhagic events. After 6 months, the recurrence of the hemorrhagic syndrome was noted, with an extensive hematoma in the left genian region, multiple disseminated bruises on the trunk, upper and lower limbs, and increased APTT. The corticosteroid treatment, FEIBA, and eptacog were resumed, leading to the remission of the hemorrhagic syndrome. The hemorrhagic events were controlled under periodic treatment with eptacog alfa and FEIBA until 5 months; then, an extensive bruise was observed on the left calf and the dorsal side of the left foot with an increased APTT (45.1 s). Inhibitors were measured again at 3 Bethesda units/mL. The soft tissue ultrasound revealed a secondary collection suggestive of a hematoma in the anterior tibial muscle with an approximate diameter of 10/20/80 mm (AP/T/CC), with a diffuse edematous-infiltrated appearance, an apparently inactive Baker’s cyst, and the diffuse hyperechoic infiltration of the subcutaneous adipose tissue corresponding to the anterolateral bruise of the upper portion of the left calf. Treatment with dexamethasone at 16 mg/day and eptacog alfa at a dose of 90 µg/kg was administered, with a partial improvement in symptoms. The worsening of the cutaneous-mucosal hemorrhagic syndrome was noted 2 months after (in the soft tissue ultrasound, the following was identified: at the posteroinferior aspect of the right thigh, no collections corresponding to the topography of the palpable lesion from the middle third of the thigh were visualized, without hypoechoic streaks at the interface of the adipose plane with the muscular plane of the lower thigh of the upper calf; a hypoechoic collection at the knee joint was more voluminous at the suprapatellar level), which led to the repetition of inhibitor tests (increased values >50 Bethesda units) and the addition of imatinib at 100 mg/day to the treatment. The clinical evolution was slowly favorable under treatment with eptacog alfa and imatinib at 50 mg/day, with the slow resorption of hematoma collections and pain control under analgesic treatment. After 6 months, the absence of factor VIII inhibitors and a normal coagulation profile were noted, with no other hemorrhagic episode that required surgical supervision.

#### 2.1.2. Case 2

A 33-year-old patient with a history of severe type A hemophilia, post-hemorrhagic stroke, and multiple hemophilic arthropathies, presenting with the semi-ankylosis of both knees fixed in semiflexion, was admitted for a bleeding tumor formation on the second toe of the right foot. The paraclinical test revealed inflammatory syndrome (Table 2).

The surgical consultation established the diagnosis of toe II gangrene on the level of the right foot. An X-ray taken at the level of the right forefoot (front and profile) described intense systemic osteoporosis, halus valgus, and the absence of radiological signs of osteitis with recommendations for surgical intervention.

Substitution treatment with coagulation factor VIII was administered every 12 h. Surgery was performed by practicing the amputation of the second toe of the right foot with the resection of the second metatarsal. The patient was transferred to the Hematology Clinic for further specialized treatment, with a relatively good general condition. Substitution treatment with coagulation factor VIII was administered at 12 h with hemostatic treatment with a favorable clinical–biological evolution. Under local antiseptic cleansing, the evolution was favorable with the healing of the amputation stump. The patient was discharged after 11 days of hospitalization.

#### 2.1.3. Case 3

A 21-year-old patient with a history of hemophilia type A—severe form (F VIIIc < 1%)—from the age of 8 months presented to the emergency room for an affected general condition and diffuse abdominal pain. Clinical and paraclinical explorations (Table 3) established the diagnosis of a ruptured retroperitoneal hematoma in the peritoneal cavity with hemoperitoneum.

Exploratory laparotomy with peritoneal lavage was performed. Substitution treatment with coagulation factor VIII and von Willebrand Factor 40 UI/kgbw was administered concurrently with tranexamic acid. During hospitalization, the administration of substitution treatment with coagulation factor VIII and the von Willebrand Factor for 8 h was continued simultaneously with transfusions of fresh frozen plasma until the arrival of the result of the dosing of inhibitors with a positive result (inhibitor titer of 573.4 Bethesda units/mL), when replacement therapy was substituted with eptacog alfa (90 µg/kgbw). Injectable corticoid, antibiotic, hemostatic, and antalgic treatment was also administered, with a favorable clinical–biological evolution.

The control ultrasound confirmed a gradual reduction in the previously described fluid collection alongside a favorable clinical and biological recovery, leading to the patient’s discharge approximately 30 days after the surgery.

He returned as an emergency about 30 days after discharge with an affected general condition and diffuse abdominal pain. An abdominal ultrasound revealed the left kidney to be without pyelocaliceal dilatation inferior to it, partially masking the pole and with the inferior left renal inhomogeneous collection of 110/115/205 mm. The abdominal–pelvic CT, both native and with a contrasting substance, objectifies left iliopsoas retroperitoneal hematoma with inguinal extension and the absence of post-traumatic bone lesions in the pelvis. Eptacog alfa substitution treatment (90 µg/kgbw at 2 h), injectable corticoid treatment, cyclophosphamide treatment, hemostatic, and hepatoprotective were administered with favorable clinical–biological evolution.

The evolution was favorable with complete resorption of the retroperitoneal hematoma 7 months post-procedure, during which period he was monitored monthly and hemostatic substitution treatment was administered.

He returned 5 months later in emergency mode for a serious general condition: he was hemodynamically stable with the suspicion of hemorrhagic shock and the hematoma of the iliopsoas muscle. He was monitored daily from a hematological point of view with the recommendation of hemostatic doses and the dynamic follow-up of the coagulation profile due to the patient’s vital risk and reserved prognosis.

Replacement hemostatic treatment with a coagulation factor, eptacog alfa (90 µg/kgbw), etamsilat, vitamin K, carbazochrome, tranexamic acid, and immunosuppressive treatment (imatinib, dexamethasone) was administered.

An abdominal ultrasound objectifies the right retroperitoneal inhomogeneous fluid collection measuring approximately 200/180/110 mm, similar left retroperitoneal 200 mm thick collection on mesenteric root topography, the visibility of the bilateral pyelocaliceal system, transonic intraperitoneal fluid visible in all spaces 200 mm thick subhepatic, and 70 mm in the hypogastrium.

During hospitalization, the administration of substitution treatment with coagulation factor eptacog alfa (90 µg/kgbw), etamsilat, vitamin K, and carbazochrome continued. Also, injectable corticoid treatment was administered with a favorable clinical–biological evolution.

The ultrasound monitoring objectified the gradual decrease in the size of the collections and also a decrease in the intraperitoneal fluid. The patient was discharged after 12 days of hospitalization.

The patient was clinically and biologically monitored monthly with the administration of hemostatic substitution treatment with favorable evolution.

#### 2.1.4. Case 4

A 52-year-old male patient, known to have hemophilia A (factor VIII activity < 1%) with inhibitors, was hospitalized due to three days of worsening upper abdominal pain. The patient’s medical history included hemophilic arthropathy and chronic hepatitis B and C from past plasma transfusions. Clinical examination revealed pain on palpation in the upper right quadrant with no evidence of peritoneal irritation or a positive Murphy’s sign that suggested acute cholecystitis, and the laboratory test revealed inflammatory syndrome. (Table 4).

The ultrasound performed showed an enlarged gallbladder with stones. We initiated treatment with antibiotics with Ampicillin at 2 g/8 h, antispasmodics, and hemostatic support with eptacog alfa (90 µg/kgbw) at 12 h. After four days, laparoscopic cholecystectomy was planned, considering the hemorrhagic risk. During surgery, a pericholecystic plastron with colon adhesion was found, leading to the conversion to a right subcostal incision (Figure 1). Despite careful dissection, severe tissue inflammation altered the subhepatic anatomy. The surgery lasted 3 h with a blood loss of approximately 150 mL.

Continuous eptacog alfa administration was maintained, but after 10 h, the patient exhibited signs of hypovolemia. Emergency surgical reintervention revealed intraperitoneal bleeding and diffuse hemorrhage; perihepatic packing was performed. Postoperatively, eptacog alfa was administered every 2 h to maintain factor levels. The patient’s hemoglobin stabilized, and on the seventh day after reintervention, hepatic packing was removed. Polymyxin E and Linezolid were prescribed for wound infection as the sample from the abdominal wall secretion was antibiogram-positive for Acinetobacter baumanii and Enterococus faecium. Hemostatic treatment with eptacog alfa continued in doses of 90 μg/kgbw every 6–12 h until discharge on day 23 postoperatively. The patient, free of hemorrhagic events, was discharged with a hemoglobin level of 11 g/dL. The patient was discharged after 31 days of hospitalization.

#### 2.1.5. Case 5

A 46-year-old patient with a medical history of hemophilia type A, chronic hepatitis C, epilepsy, and arthropathy presented to the emergency department suffering from diffuse abdominal pain and in a serious overall condition. The abdominal Computed Tomography objective showed loops of the small intestine dilated up to a caliber of 34 mm, with a stratified wall, with contrast enhancement, and a thickness of up to 5 mm. The colonic frame had a stratified wall, with contrast enhancement, and a thickness of 10 mm at the level of the rectum. Transverse colon and hepatic flexure with minimal contrast enhancement and a wall thickness of up to 15 m was determined with the conclusion of colitis with ischemia at the level of the transverse colon. The paraclinical examination revealed inflammatory syndrome (Table 5).

Eptacog alfa treatment was initiated, and an exploratory laparotomy was conducted as a surgical intervention. This procedure revealed ischemic pancolitis, leading to the performance of a total colectomy with the partial resection of the rectum (Figure 2).

The evolution was slowly favorable under treatment with eptacog alfa at 90 µg/kgbw and discharge after 25 days postoperatively, with monitoring at another center.

## 3. Discussion

Hemophilia patients require continuous monitoring due to the recurrent nature of the hemorrhagic risk, even under replacement therapy. This recurrent characteristic can be observed in cases 1 and 3, with these patients being hospitalized on multiple occasions for the recurrence of hemorrhagic lesions. However, in most cases, hemorrhagic recurrence responds to conservative treatment, and emergency surgical intervention is not necessary.

Surgical indications appear when all other conservative means of treatment have failed. In the case of patients with hemophilia, the ideal indication is for conservative treatment, considering the increased additional risk of an intra- and postoperative hemorrhage, even with properly administered hemostatic treatment. Conservative treatment cannot be instituted in cases with an acute surgical abdomen, with the only solution being emergency surgery. If conservative treatment fails, surgical treatment should be considered. Surgery for patients with hemophilia requires planning additional and continuous interaction between support team members medically versus what is needed for other patients.

In instances of emergency bleeding, such as trauma or urgent surgical procedures, prompt intervention is imperative. Immediate action involves administering a significant dosage of factor VIII concentrate to attain hemostasis. Consequently, surgeries for individuals with hemophilia A should be carried out in a facility equipped with adequate laboratory capabilities for evaluating factor VIII levels throughout the perioperative period, including inhibitor screening. Furthermore, optimal support from the blood bank is essential to ensure an ample provision of factor VIII concentrates [6]. In our case, the necessary amount of treatment was available in all cases through the Romanian National Program For The Treatment Of Hemophilia and Thalassemia that takes place within the Hematology Clinic in Iasi.

All currently marketed plasma-derived or recombinant factor VIII coagulation factor concentrates are listed in the World Health Organization Online Registry of Coagulation Factor Concentrates [7]. The concentration of different types of coagulation factors expressed in IU varies between 250 and 3000 IU. In the absence of inhibitors, each IU of plasma or recombinant short half-life factor VIII administered intravenously per kilogram of body weight increases the plasma level of factor VIII by approximately 2 IU/dL.

This growth is dependent on a number of individual factors, the most important of which is body mass index (BMI). The half-life of factor VIII, with a short-half-life, is approximately 12 h in adults; it is shorter in children and increases with age. The formula for calculating the dose to be administered is dependent on the patient’s weight in kilograms and the desired level of FVIII in IU/dL. Their product is multiplied by 0.5 [1,8]. The calculation of factor VIII replacement for bleeding in severe hemophilia A is as follows:Dose of factor VIII = percentage desired of factor × bodyweight (kg) × 0.5

For severe hemorrhages, it is recommended to administer factor VIII in order to achieve a 100% factor VIII level, and for mild to moderate hemorrhage, it is recommended to achieve a 30% to 50% factor VIII level [9].

In general, surgical procedures in people with hemophilia are traditionally classified as major or minor, but procedures that may be designated “minor” for the general population may be considered “major” for people with hemophilia, where other associated factors may alter the classification (disease severity, presence of inhibitors, and other comorbidities). The World Federation of Hemophilia defines major surgery as the procedure requiring the penetration of cavities, mesenchymal barriers or fasciae, the surgical removal of an organ, or anatomical surgical reconstruction. Minor surgery is defined as a procedure that requires the manipulation of the skin, mucous membranes, and superficial connective tissue, catheter/port-a-cath implantation, or cutaneous excision [4].

In the surgical literature, interventions that are conducted under general anesthesia are categorized as major surgical procedures, whereas those typically performed using local anesthesia are considered minor surgical procedures.

This classification is useful for predicting the risk of bleeding associated with surgical procedures, assessing plasma factor levels and inhibitor status to plan for factor replacement, ensuring optimal intraoperative and postoperative conditions, and guaranteeing the availability of necessary staff based on the extent of surgical intervention. This includes having an experienced surgeon, hematologist, pharmacist and continuous monitoring through laboratory tests [10]. Unfortunately, our center does not have the necessary infrastructure for inhibitor assays; therefore, in case 4, we administered treatment without these values. The tests were processed through a private laboratory in another country, with the final result available in 10 days.

Another requirement is interdisciplinary management by surgeons, hematologists, and anesthesiologists who have experience with the treatment of hemophilic patients. It is crucial to maintain hemodynamic conditions as closely as possible to the normal range during anesthesia, as hypertension and tachycardia can lead to heightened bleeding at the surgical site. When performing tracheal extubating, it is essential to do so in a deep plane of anesthesia, ensuring the absence of the cough reflex and exercising extreme vigilance to prevent pharyngeal aspiration [11].

The medical management of patients like this requires a personalized approach, selecting the appropriate combination of available therapies for each individual case. Another issue is the development of inhibitors in hemophilic patients, which complicates the management of bleeding events, as standard replacement approaches are no longer effective [12]. All patients with hemophilia should also be screened for the presence of inhibitors to factor VIII, and their prior factor exposure and inhibitor history should be reviewed, as the presence of an inhibitor has an impact on hemostatic treatment options [13].

In case number 4, the patient, who was known to have a severe form of hemophilia with inhibitors and was a high responder, was being treated with activated recombinant alpha eptacog. It was essential to provide adequate replacement therapy with the coagulation factor consistently, spanning hospitalization, the intraoperative phase, and post-intervention to ensure a positive outcome for this patient.

When using bypassing agents in the management of hemophilia patients with inhibitors, the routine assessment of factor levels or activated partial thromboplastin time (aPTT)/prothrombin time (PT) is not effective for monitoring therapy. This is because bypassing agents, such as eptagos alfa, work by circumventing the traditional pathways assessed by these tests, making them unreliable indicators of therapeutic efficacy. Therefore, alternative parameters are needed to guide the choice of bypassing agent, optimize dosing, and monitor the response to therapy [14].

Viscoelastic testing, such as thromboelastography (TEG) or rotational thromboelastometry (ROTEM), may provide useful information for guiding the use of bypassing agents. These tests offer a more comprehensive assessment of the clotting process by measuring the kinetics, strength, and stability of clot formation. Occasionally, patients demonstrate clinical responsiveness to either aPCC or rFVIIa, and viscoelastic tests have been used to determine the agent to which the patient shows a better response. This approach can be particularly valuable in scenarios of uncontrolled bleeding or perioperative planning in patients with inhibitors, allowing for a more targeted and effective management strategy [15]. Unfortunately, our university center, despite its surgical capabilities, does not have the necessary equipment to perform a thromboelastographic examination.

From the point of view of the abdominal wall approach, the evidence currently suggests that skin incisions using electrosurgery are faster and associated with less blood loss than those performed with a classic scalpel, and there are no differences in the rate of wound complications or postoperative pain. Conventional surgery prefers meticulous hemostasis using ligations for all visible bleeding and cautery only for capillary bleeding while trying to keep the operative time to a minimum.

Specialized research indicates that direct current electrocautery is a dependable and secure method for long-term hemostasis in surgical procedures involving bleeding disorders. Moreover, it offers the advantage of being cost-effective while maintaining comparable rates of complications [16,17]. Unfortunately, our university center does not have the necessary equipment to perform thromboelastography.

Laparoscopic interventions are recommended to minimize damage to the abdominal wall using access ports. Hospital stays are significantly shorter for patients undergoing laparoscopic procedures compared to those receiving traditional surgery, leading to lower therapeutic costs and quicker patient mobilization. Conversely, the length of postoperative hospital stays for patients with hemophilia tends to be longer than those for non-hemophilic patients due to the necessity for ongoing substitution therapy administration [18,19].

The management of the patient with hemophilia A who required major surgery was conducted in our university center by the “Guide to Replacement Therapy in Hemophilia under Normal Resource Conditions”, which respects The World Federation of Hemophilia recommendations. In hemophilia A with inhibitors, we administrated eptacog alfa at 90 µg/kg right before the surgical intervention, with doses at every 2 h for the first 24–48 h after the intervention. This dose was kept at every 2 h in the first 6–7 days, after which we raised the time to 6–8 h for a minimum of 14 days. The administration was performed intravenously in bolus for 2–5 min. In hemophilia A without inhibitors, we administered factor VIII at 8–12 h in order to maintain a level of 80–100% of the factor activity until the healing of the incisional site, after which we maintained the administration for 10–14 days to obtain a 30–60% level of factor activity [4].

## 4. Conclusions

The timing of surgical intervention is very important, so the indication for surgical intervention must be made judiciously and without delay. Therefore, it is crucial to have access to hemostatic support in order to carry out surgery on these patients, which is essential for potentially life-saving interventions.

Performing surgery on patients with hemophilia necessitates a center equipped with appropriate therapies, a team experienced in multidisciplinary approaches, a comprehensive treatment plan for hemostasis, and a detailed surgical strategy. This plan must account for every potential eventuality that could occur during the surgery.

For optimal outcomes in surgical interventions for hemophilic patients, it is necessary to ensure adequate preoperative and postoperative monitoring of coagulation factors, meticulous hemostasis during surgery, and careful postoperative care, including the timely administration of replacement therapy. Postoperative bleeding complications are rare and can typically be managed with conservative treatment.

In the cases we presented, we attempted to provide the most appropriate treatment by mobilizing all necessary resources through multidisciplinary and multicenter collaboration to achieve a favorable outcome in each case.

We encountered difficulties due to the absence of a laboratory for inhibitor testing, limited quantities of hemostatic treatment, and the lack of multiple teams specialized in the management of hemophilic patients.

## Figures and Tables

**Figure 1 healthcare-12-00610-f001:**
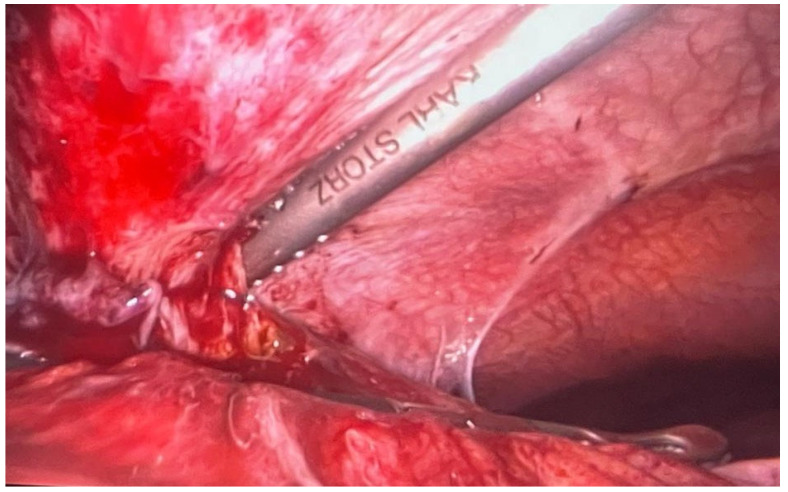
Pericholecystic plastron.

**Figure 2 healthcare-12-00610-f002:**
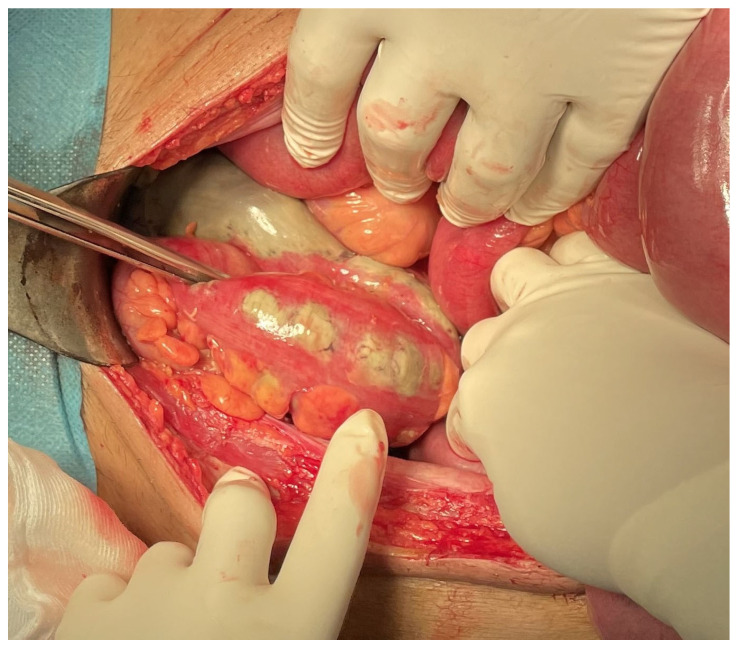
Ischemia of the colon.

**Table 1 healthcare-12-00610-t001:** Case 1—laboratory parameters.

Clinical Examination and Treatment				
Symptoms	Diagnostic	Surgical Intervention	Hemostatic Treatment	Evolution
Edema and pain in the calf and right leg, chills, and fever	A superinfected hematoma	Puncture and evacuation	Complex anti-inhibitors of factor VIII activity (FEIBA)	Favorable
Paraclinical examination				
Parameter	Value	Parameter	Value	
Leukocytes	12,370/µL	Fibrinogen	567 mg/dL	
Neutrophils	95,000/µL	INR	1.02	
Hemoglobin	12.6 g/dL	PT	12.3 s	
Hematocrit	38.9%	aPTT	39.1 s	
RBC	4.33 × 10^6^/µL	aPTT (%)	1.45	
MCV	89.8 fL	Prothrombin activity	97%	
MCH	29.1 pg	FDPs	++	
MCHC	32.4 g/dL	Inhibitors	-	
Thrombocyte	228,000/µL			

**Table 2 healthcare-12-00610-t002:** Case 2—laboratory parameters.

Clinical Examination and Treatment				
Symptoms	Diagnostic	Surgical Intervention	Hemostatic Treatment	Evolution
Bleeding from tumoral formation at the level of the second toe of the right foot	Toe II gangrene on the level of the right foot	Amputation of the second toe of the right foot with resection of the second metatarsal.	Coagulation factor VIII	Favorable
Paraclinical examination				
Parameter	Value	Parameter	Value	
Leukocytes	8790/µL	Fibrinogen	387 mg/dL	
Neutrophils	82,400/µL	INR	1.04	
Hemoglobin	12.9 g/dL	PT	12.3 s	
Hematocrit	40.1%	aPTT	48.8 s	
RBC	5.23 × 10^6^/µL	aPTT (%)	1.81	
MCV	76.7 fL	Prothrombin activity	104%	
MCH	24.7 pg	FDPs	-	
MCHC	32.2 g/dL	Inhibitors	-	
Thrombocyte	294,000/µL			

**Table 3 healthcare-12-00610-t003:** Case 3—laboratory parameters.

Clinical Examination and Treatment				
Symptoms	Diagnostic	Surgical Intervention	Hemostatic Treatment	Evolution
Affected general condition and diffuse abdominal pain	Giant ruptured retroperitoneal with hemoperitoneum	Exploratory laparotomy with peritoneal lavage	Coagulation factor VIII and von Willebrand Factor	Favorable
Paraclinical examination				
Parameter	Value	Parameter	Value	
Leukocytes	25,060/µL	Fibrinogen	472 mg/dL	
Neutrophils	19,000/µL	INR	1.77	
Hemoglobin	8.6 g/dL	PT	19.5 s	
Hematocrit	25.7%	aPTT	109.1 s	
RBC	3.43 × 10^6^/µL	aPTT (%)	4.04	
MCV	74.9 fL	Prothrombin activity	49%	
MCH	25.1 pg	Inhibitors	+	
MCHC	33.5 g/dL			
Thrombocyte	631,000/µL			

**Table 4 healthcare-12-00610-t004:** Case 4—laboratory parameters.

Clinical Examination and Treatment				
Symptoms	Diagnostic	Surgical Intervention	Hemostatic Treatment	Evolution
Upper abdominal pain	Acute cholecystitis	Classic cholecystectomy	Eptacog alfa	Favorable
Paraclinical examination				
Parameter	Value	Parameter	Value	
Leukocytes	14,030/µL	Fibrinogen	598 mg/dL	
Neutrophils	10,480/µL	CRP	9.53 mg/dL	
Hemoglobin	15.1 g/dL	INR	14.90	
Hematocrit	44.6%	PT	12.3 s	
RBC	5.27 × 10^6^/µL	aPTT	96.6 s	
MCV	84.6 fL	aPTT (%)	3.58	
MCH	28.7 pg	Prothrombin activity	68%	
MCHC	33.9 g/dL	Inhibitors	+	
Thrombocyte	474,000/µL			

**Table 5 healthcare-12-00610-t005:** Case 5—laboratory parameters.

Clinical Examination and Treatment				
Symptoms	Diagnostic	Surgical Intervention	Hemostatic Treatment	Evolution
Affected general condition and diffuse abdominal pain	Ischemic pancolitis	Total colectomy with partial resection of the rectum	Eptacog alfa	Favorable
Paraclinical examination				
Parameter	Value	Parameter	Value	
Leukocytes	13,020/µL	Fibrinogen	695 mg/dL	
Neutrophils	89,000/µL	CRP	39.98	
Hemoglobin	15.7 g/dL	INR	1.47	
Hematocrit	44.2%	PT	16.2 s	
RBC	5.34 × 10^6^/µL	aPTT	38.3 s	
MCV	82.8 fL	aPTT (%)	1.45	
MCH	29.4 pg	Prothrombin activity	64%	
MCHC	35.5 g/dL			
Thrombocyte	204,000/µL			

## Data Availability

All data generated or analyzed are included in this case report. Further enquiries can be directed to the corresponding author.

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
