# Peer review of "Surgical Emergencies in Patients with Hemophilia A—What to Expect"

_healthcare, 2024, doi:10.3390/healthcare12060610_

Round 1
Reviewer 1 Report
Comments and Suggestions for Authors
Adelina et al., have reported several cases of hemophilia A and their handling. The cases are well reported and handled accordingly. The description, treatment and usage of coagulation Factor VIII and vice versa was logical.
This reviewer has a few questions?
Can you please mention post-op care?
Did you think about combination therapy?
Several antibiotics affect the treatment regime concerning coagulation especially in autoimmune diseases. Can you please specify your antibiotics in various cases?
I would suggest you please summarize your cases into a table showing their clinical signs presented, diagnosis, treatment/surgical intervention and post op etc. along with their prognosis. It would really enhance the presentation quality of your work.
Author Response
Thank you for the time and effort you dedicated to reviewing our article. Your insightful feedback and constructive suggestions have been valuable in enhancing the quality and clarity of our work.

Reviewer 2 Report
Comments and Suggestions for Authors
Thank you for the opportunity to review the manuscript.
- English needs extensive improvement
- A table summarizing patients characteristics is required
- A table with laboratory parameters is needed
- Was TEG performed in any of the patients? If yes, it could be beneficial to add it to the manuscript. If not, a description and graph on how the TEG could appear in patients with hemophilia could improve the manuscript
- a discussion/summary about the guidelines/good clinical practice would be beneficial
- Line 45: provide reference
Comments on the Quality of English LanguageEnglish needs extensive improvement
Author Response

(The authors gave the same response as above.)

Reviewer 3 Report
Comments and Suggestions for Authors
The authors are presenting their valuable experience in the field of surgical emergencies in patients with haemophilia, a field confronted with numerous risks, difficulties and obstacles: it is about life- saving provocative interventions .
Unfortunately , their article is not reflecting in an appropriate manner the subject, claiming a lot of major and minor changes. I will mention in the following only the most important corrections to be performed:·
- Many failure in the use of English language (sentences without predicate, non-concordance between subject and predicate, grammar errors, strange words :1/35, 2/60-63, 2/71, 2/69, 4/153, 3/111, 4/172,5/245, 5/251-252, 5/249...) and careless drafting (2-3 times repeated word in a sentence 2/83, 2/60-69), inadequate expressions (4/201, 5/213, 5/235 , 5/251)
· -Lack of the main subject on the list of keywords (surgical emergencies)
· - Introduction not focused on the objectives of the presentation ( definition of surgical emergencies, major and minor interventions , main risks...),being presented basic, generally known data regarding the content of FVIII UI / vial, pharmaco-kinetic data and modality of calculation of the dosage of factor concentrate (1/45-46 , 2/47-59)
· -The case presentations are performed in a narrative style
*without documented diagnosis (without pre-surgical plasma FVIII level, inhibitors status..)(case 1,2,3,5); the meaning of the abbreviation RR (row 147)
*treatment expressed in vials of factor concentrate, not in IU ( case 2,case 3), mostly without details regarding pro-duct, dosage and its changes along the surgical treatment; or an inappropriate statement: factor performed from hospitalization (2/64)
*lack of monitoring program and its results
*medically incorrect date (2/75, 2/79, 4/169, 4/171 ) and lack of justification of some treatments, like corticoids, vitamin K, immuno-suppressive treatment, antispasmodic (case 2 )
*unanswered questions *in case 4 with inhibitors the therapy with eptacog was stopped and resumed on postoperative day 8 ; when and why? (4/165)
*in case 3 (haemophilia A with inhibitors ) there have been used more than 100 vials of FVIII+Willebrand factor and 8 units fresh frozen plasma …until the arrival of the results of inhibitors evaluation…how long took it in this emergency condition...where has it been performed?(3/103-108)
- The discussions are without some comparable data from the literature
- The conclusions should be focused on the own experience ...difficulties met …challenges.
Comments on the Quality of English Language
The authors are presenting their valuable experience in the field of surgical emergencies in patients with haemophilia, a field confronted with numerous risks, difficulties and obstacles: it is about life- saving provocative interventions .
Unfortunately , their article is not reflecting in an appropriate manner the subject, claiming a lot of major and minor changes. I will mention in the following only the most important corrections to be performed:·
- Many failure in the use of English language (sentences without predicate, non-concordance between subject and predicate, grammar errors, strange words :1/35, 2/60-63, 2/71, 2/69, 4/153, 3/111, 4/172,5/245, 5/251-252, 5/249...) and careless drafting (2-3 times repeated word in a sentence 2/83, 2/60-69), inadequate expressions (4/201, 5/213, 5/235 , 5/251)
· -Lack of the main subject on the list of keywords (surgical emergencies)
· - Introduction not focused on the objectives of the presentation ( definition of surgical emergencies, major and minor interventions , main risks...),being presented basic, generally known data regarding the content of FVIII UI / vial, pharmaco-kinetic data and modality of calculation of the dosage of factor concentrate (1/45-46 , 2/47-59)
· -The case presentations are performed in a narrative style
*without documented diagnosis (without pre-surgical plasma FVIII level, inhibitors status..)(case 1,2,3,5); the meaning of the abbreviation RR (row 147)
*treatment expressed in vials of factor concentrate, not in IU ( case 2,case 3), mostly without details regarding pro-duct, dosage and its changes along the surgical treatment; or an inappropriate statement: factor performed from hospitalization (2/64)
*lack of monitoring program and its results
*medically incorrect date (2/75, 2/79, 4/169, 4/171 ) and lack of justification of some treatments, like corticoids, vitamin K, immuno-suppressive treatment, antispasmodic (case 2 )
*unanswered questions *in case 4 with inhibitors the therapy with eptacog was stopped and resumed on postoperative day 8 ; when and why? (4/165)
*in case 3 (haemophilia A with inhibitors ) there have been used more than 100 vials of FVIII+Willebrand factor and 8 units fresh frozen plasma …until the arrival of the results of inhibitors evaluation…how long took it in this emergency condition...where has it been performed?(3/103-108)
- The discussions are without some comparable data from the literature
- The conclusions should be focused on the own experience ...difficulties met …challenges.
Author Response

(The authors gave the same response as above.)

Round 2
Reviewer 2 Report
Comments and Suggestions for Authors
Thank you for addressing the comments
Comments on the Quality of English Language.
Author Response
Thank you for the time allocated to the review of our article and also for the suggestions that were of real use in improving the work.
